# Transcutaneous Cervical Spinal Cord Stimulation Combined with Robotic Exoskeleton Rehabilitation for the Upper Limbs in Subjects with Cervical SCI: Clinical Trial

**DOI:** 10.3390/biomedicines11020589

**Published:** 2023-02-16

**Authors:** Loreto García-Alén, Hatice Kumru, Yolanda Castillo-Escario, Jesús Benito-Penalva, Josep Medina-Casanovas, Yury P. Gerasimenko, Victor Reggie Edgerton, Guillermo García-Alías, Joan Vidal

**Affiliations:** 1Fundación Institut Guttmann, Institut Universitari de Neurorrehabilitació Adscrit a la UAB, 08916 Badalona, Spain; 2Universitat Autónoma de Barcelona, 08193 Barcelona, Spain; 3Fundació Institut d’Investigació en Ciéncies de la Salut Germans Trias i Pujol, 08916 Badalona, Spain; 4Institute for Bioengineering of Catalonia, Barcelona Institute of Science and Technology, 08028 Barcelona, Spain; 5Department of Automatic Control, Universitat Politécnica de Catalunya-Barcelona Tech (UPC), 08028 Barcelona, Spain; 6Centro de Investigación Biomédica en Red de Bioingeniería, Biomateriales y Nanomedicina, 28029 Madrid, Spain; 7Pavlov Institute of Physiology, St. Petersburg 199034, Russia; 8Department of Physiology and Biophysics, University of Louisville, Louisville, KY 40292, USA; 9Kentucky Spinal Cord Injury Research Center, University of Louisville, Louisville, KY 40292, USA; 10Rancho Research Institute, Los Amigos National Rehabilitation Center, Downey, CA 90242, USA; 11Departament de Biologia Cel·lular, Fisiologia i Immunologia & Insititute of Neuroscience, Universitat Autónoma de Barcelona, Bellaterra, 08193 Barcelona, Spain

**Keywords:** transcutaneous electrical spinal cord stimulation, cervical spinal cord injury, upper extremity, robotics, functionality, motor function, grip force

## Abstract

(1) Background: Restoring arm and hand function is a priority for individuals with cervical spinal cord injury (cSCI) for independence and quality of life. Transcutaneous spinal cord stimulation (tSCS) promotes the upper extremity (UE) motor function when applied at the cervical region. The aim of the study was to determine the effects of cervical tSCS, combined with an exoskeleton, on motor strength and functionality of UE in subjects with cSCI. (2) Methods: twenty-two subjects participated in the randomized mix of parallel-group and crossover clinical trial, consisting of an intervention group (*n* = 15; tSCS exoskeleton) and a control group (*n* = 14; exoskeleton). The assessment was carried out at baseline, after the last session, and two weeks after the last session. We assessed graded redefined assessment of strength, sensibility, and prehension (GRASSP), box and block test (BBT), spinal cord independence measure III (SCIM-III), maximal voluntary contraction (MVC), ASIA impairment scale (AIS), and WhoQol-Bref; (3) Results: GRASSP, BBT, SCIM III, cylindrical grip force and AIS motor score showed significant improvement in both groups (*p* ≤ 0.05), however, it was significantly higher in the intervention group than the control group for GRASSP strength, and GRASSP prehension ability (*p* ≤ 0.05); (4) Conclusion: our findings show potential advantages of the combination of cervical tSCS with an exoskeleton to optimize the outcome for UE.

## 1. Introduction

A spinal cord injury (SCI) disrupts information between the supraspinal centers and muscles, leading to varying degrees of paralysis that greatly impact functional ability [1]. Approximately 50% of subjects affected by spinal cord injury present a lesion in the cervical region [2], so there is a high percentage of subjects with severe limitations in the execution of basic activities of daily life [2]. Thus, restoring arm and hand function and re-gaining greater independence in performing daily activities is a very high priority for individuals with tetraplegia [3]. 

Recently electrical stimulation of the spinal cord either via implanted (eSCS) or transcutaneous (tSCS) electrodes has emerged as a viable neuromodulation approach for facilitating the recovery of motor function in subjects with spinal cord injuries (SCI) [4]. In 1973, for the first time, it was reported that there was an improved motor activity during eSCS in pain study in subjects with multiple sclerosis [5]. Following this initial observation, the technique was studied in other neurological diseases, such as SCI. Using eSCS in SCI, there were reported improvements in voluntary movements of the upper [6] and lower extremities [7,8,9,10,11,12,13,14,15], improvement in cardiovascular [16], bladder and bowel function [17,18,19], and reduction of spasticity [20,21,22]. On the other side, previous studies demonstrated that tSCS is also a promising technique to improve the voluntary motor activity of the upper limbs [23,24,25,26,27], and lower limbs [28,29,30,31,32,33,34,35], to improve trunk stability [36,37], to reduce spasticity [38,39,40], improve lower urinary tract [41,42], and cardiovascular function [43,44].

As mentioned above, achieving total or even partial recovery of hand function is one of the main objectives of a rehabilitation program following cervical spinal cord injury [3]. Studies applied tSCS at cervical segments combined with activity-based upper limb rehabilitation such as standard stretches, active-assisted range of motion exercises, gross and fine motor skill training, maximum voluntary contraction, and unimanual and bimanual task performance have shown significant improvements in upper limb function [23,25,26,27]. 

There are few studies that explore the effect of the combination of tSCS with hand training in subjects with cervical spinal cord injury (cSCI). The majority of them are case studies or clinical trials with small sample sizes and without a control group. Only one crossover study compared intensive functional task training and combination with tSCS in six subjects with chronic cSCI [26]. The improvement, however, was significantly higher for the intervention than for the control group. Using tSCS combined with activity-based rehabilitation, functional changes emerge more rapidly and to a greater degree than in insolation because the activity may help to enhance neuroplasticity. It is noteworthy also to mention here that tSCS combined with hand training enhanced the grip strength in healthy subjects when compared to hand training or tSCS applied alone [45].

So, the effects of the tSCS technique for upper extremity rehabilitation are understudied and there is a need for randomized controlled clinical trials with a bigger number of subjects with cSCI [23,24,25,26,27]. 

Our aim was to study the effects of cervical tSCS combined with rehabilitation assisted by a robotic exoskeleton on motor strength and functionality of the upper extremity in a large number of cSCI subjects. The reason for having used a robotic exoskeleton was that provides the potential for an intensive training regimen that can provide a wide range of volume and repetitive movements that can be administered accurately, coherently, and physiologically compatible [46,47]. Our hypothesis was that subjects who receive tSCS combined with rehabilitation assisted by an upper limb robotic exoskeleton could improve motor strength and functionality more than those who received a robotic exoskeleton alone.

## 2. Materials and Methods

### 2.1. Subjects 

The participants were recruited in the functional rehabilitation program of the Institut Guttmann (Badalona, Spain). Inclusion criteria were: (i) male or female, more than 18 years old; (ii) a stable traumatic or no traumatic cervical SCI; (iii) time since SCI: 3–12 months; (iv) AIS A, B, C, and D with but at least ≥ 2 muscles affected in the upper extremities, with a total sum of muscle strength more than 2 points in two muscles (the wrist extension together with abductor pollicis brevis (APB) at least in one side). Exclusion criteria were: (i) unstable medical condition (cancer, acute infections, etc.); (ii) dependent on mechanical ventilation; (iii) severe spasticity (≥3 score on the Modified Ashworth scale–MAS), contraindication for rehabilitation assisted with a robotic exoskeleton (Armeo^®^Power); (iv) peripheral nerve injury; (v) intolerance of tSCS, peacemakers, electronic implants, or episodes of epilepsy; (vi) participating in another investigation. 

The protocol was approved by the Ethics Committee of the Institut Guttmann and was carried out in accordance with the standards of the Declaration of Helsinki. All subjects were informed of all experimental procedures, after which each subject completed a signed informed consent.

### 2.2. Experimental Design

The experimental design at the beginning was a randomized, controlled clinical trial, which consisted of two groups: (i) intervention group: tSCS combined with a robotic exoskeleton (Armeo^®^Power), and (ii) control group: robotic exoskeleton (Armeo^®^Power) alone. We used a computer-generated list as a randomization strategy. The assignment of the subjects to the treatment interventions was random. If the subjects wanted to participate in both groups, we gave them this possibility. The duration of both interventions were 2 weeks, during which subjects performed four sessions per week. All patients from each group were evaluated at baseline condition, after the last session, and then two weeks after the last session for follow-up evaluation. The total duration of clinical and neurophysiological assessments was around 4–5 h. The clinical assessments were performed on three different days before the experiment to avoid fatigue. The study was carried out in the installations of the Guttmann Institute during twenty-two months, starting on August 2020 and ending on March 2022. One or two subjects were recruited per month.

### 2.3. Clinical Assessment

#### 2.3.1. Functionality Assessment of Upper Extremity

Graded Redefined Assessment of Strength, Sensibility, and Prehension (GRASSP), version 2 [48] was used to evaluate upper extremity functional capacity through four domains: strength, sensation, prehension ability, and prehension performance. Manual dexterity was measured by the Box and Block test (BBT) [49], which consists of the patient moving, one by one, the maximum number of blocks from one compartment of a wooden box to another, in 60 s. During GRASSP and BBT administration the participant was placed in a sitting upright position in the wheelchair or chair with back, 30° of shoulder flexion, 90° of elbow flexion, and forearm pronated. The required materials for the tasks were placed in front of the participant longitudinally along the edge of the table in the middle of the participant. 

The degree of independence in activities of daily living was evaluated by the Spinal Cord Independence Measure III (SCIM III) [50]. There are a total of 19 items on the SCIM III, which are divided into three subscales (self-care, respiration and sphincter management, and mobility). Scores are higher in subjects that require less assistance or fewer aids to complete basic activities of daily living and life support activities. 

#### 2.3.2. Maximal Voluntary Contraction 

The grip force was measured for both hands through three grip patterns (cylindrical grip, key lateral pinch, and tip-to-tip pinch) by a wireless handgrip dynamometer and pinchmeter (Biometric E-Link version 16, Newport, UK). 

Subjects performed three consecutive trials of 4 s, for each grip, with at least one minute of rest between each trial, recording the average of the three trials. The participant started in response to the evaluator’s verbal command “now”, performing the corresponding grip pattern as hard as possible. The participant seated upright against the back of the wheelchair, shoulder flexed 30 degrees and adducted in a neutral position, elbow flexed 90 degrees, forearm in the neutral position, wrist 0–30 degrees dorsiflexion, and 0–15 degrees ulnar deviated. 

#### 2.3.3. Neurological Assessment

AIS scale was used to evaluate the clinical motor and sensory deficit according to the International Standards for Neurological Classification of Spinal Cord Injury (ISNCSCI) [51]. The assessment was carried out with the participant in a supine position. 

#### 2.3.4. Quality of Life Assessment

Quality of life was measured by WhoQol-Bref [52] based on a four-domain structure: physical health, psychological, social relationships, environment, and two questions about an individual’s overall perception of their health. An interview-administered form was used since some subjects had no writing ability. 

### 2.4. Interventions

The study included two groups: the control and the intervention group. The control group consisted of only upper extremity training assisted by a robotic exoskeleton, Armeo^®^Power. The intervention group consisted of tSCS combined with upper extremity training assisted by Armeo^®^Power. All subjects of each group completed a total of 8 sessions over 2 weeks. Each session included 60 min of training with Armeo^®^Power with 30 min for each upper extremity with or without tSCS. 

#### 2.4.1. Hand Training with Armeo^®^Power Protocol

Armeo^®^Power is an upper extremity robotic exoskeleton that allows six actuated axes of movements: shoulder flexion/extension, shoulder adduction/abduction, shoulder internal/external rotation, elbow flexion/extension, forearm pronation/supination, wrist flexion/extension and an additional grip module. 

The protocol consisted of 30 min training constituted by six exercises for each upper extremity focused on functional movements patterns (4 min per each exercise and 15 s for rest between them): four exercises for training movement of open/close hand and two exercises for training reaching and grasping movements in two dimensions. The range of active motion was adjusted each day in each game for enhancing voluntary movement. During the evaluation baseline, we adjusted the degree of arm support, being 50% for a patient with a complete shoulder active range of motion, and 80% if the patient could not complete it; we also configured the amount of assistance required for the execution of activities, provided resistance when a participant could move actively during the given task and low assistance when the participant was unable to complete the task. The difficulty of the exercises was adjusted gradually depending on the progress of the subject.

#### 2.4.2. Transcutaneous Electrical Spinal Cord Stimulation

Electric stimulation was applied using the transcutaneous electrical stimulator BioStim-5 (Cosyma Inc., Moscow, Russia). Transcutaneous stimulation was delivered simultaneously at two sites of the cervical spinal cord along the midline between spinous processes C3-C4 and C6-C7, through 2 cm diameter hydrogel adhesive electrodes (axion GmbH, Hamburg, Germany) as cathodes and two 5 × 12 cm rectangular electrodes placed symmetrically over the iliac crests as anodes. The intensity of stimulation at each spinal level was set at 90% of RMT induced by single-pulse tSCS at the APB muscle of the less affected hand or of the right hand if both hands were similarly affected (range: 39–86 mA) (Table 1). As we reported previously [53], 90% of RMT of APB increased cervical spinal excitability more than the intensities of 80% or 110% of RMT in healthy subjects. The tSCS consisted of biphasic rectangular 1-ms pulses, each one filled with a carrier frequency of 10 kHz (i.e., each 1-ms pulse was composed of ten 0.1-ms biphasic rectangular pulses), at a frequency of 30 Hz. The tSCS was delivered during the training in the Armeo^®^Power, with time patterns of 30 s of stimulation followed by 60 s resting for 1 h. Before each session, the intensity of stimulation was increased gradually over a period of several minutes for the adaptation of the patient to tSCS. For safety, blood pressure and heart rate were monitored throughout all sessions.

### 2.5. Data and Statistical Analysis 

Data collection was performed for each subject after each assessment and data analysis was carried out after finishing the assessment of the last patient’s follow-up.

The variables were collected and then analysed separately from the right and left. In addition, the total scores were calculated from both the right and left sides for all clinical variables. 

Eight subjects who received the two interventions, since they had at least four or more weeks of washout period, we considered that they could be considered as different subjects for the next experimental group. For this reason, the most direct way was to compare the two groups (tSCS+ARMEO vs. ARMEO).

The change scores for each outcome (GRASSP, BBT, SCIM III, MVC, UEMS, AIS total motor score, AIS total sensitive score, and WhoQol-BREF) were calculated by subtracting baseline data from the data obtained after the last session and after another two weeks (follow-up) in each group (tSCS + Armeo^®^Power vs. Armeo^®^Power).

Statistical analyses were conducted with a commercial software package (IBM SPSS, version 13.0, SPSS Inc., Chicago, IL, USA). Data are presented as mean ± standard deviation (SD). 

The Kolmogorov-Smirnov test was used to examine the normality of distribution. The *t*-test was used for testing the differences between means in the variables following a normal distribution to compare the data of baseline condition with the data post-intervention and during follow-up (BBT, SCIM III, UEMS, AIS total motor score, and AIS total sensitive score). When the distribution of the data was not normal (GRASSP, MVC, and WhoQol-BREF), the Friedman test was used to compare multiple data (pre, post-intervention, and during follow-up) and for posthoc analysis, the Wilcoxon *t*-test was used. Mann Whitney-U test was used to compare the change in the scores between the intervention group and control group. The significance level was set as *p* < 0.05.

## 3. Results

### 3.1. Subjects Clinical and Demographic Characteristics

Forty-eight subjects affected by a cervical SCI in a sub-acute phase were assessed for eligibility to participate in this study (Figure 1). We recruited one or two subjects per month, who were included and randomly placed in one of the two experimental groups. Twenty-two subjects completed the inclusion criteria and signed the consent form. One of them did not finish the intervention due to shoulder pain. Eight subjects realized both interventions, three of them started in the intervention group and five started in the control group. The washout period had a mean of 46.4 ± 21.6 days (range: 30–79 days) in five SCI subjects who switched from the control group to the intervention group. Meanwhile, in three subjects who switched from the intervention group to the control group, the washout period had a mean of 70.7 ± 35.9 days (range: 42–111 days). Fifteen subjects (14 male and 1 female; SCI AIS-A/B/C/D: 2/3/4/6) formed the intervention group while the remaining fourteen (12 male and 2 female; SCI AIS-A/B/C/D:2/5/5/2) was assigned to the control group. Five follow-up evaluations were lost because of COVID isolation (they tested COVID positive and were discharged from the hospital for isolation at home) (Figure 1). 

Table 1 summarizes the demographics and clinical characteristics of the subjects. Both groups were similar with respect to age, gender, neurological level and severity of SCI, total motor score in the upper extremity, and time since SCI (*p* > 0.05 for all the comparisons, Table 1).

The mean age of subjects was 37.4 ± 13.3 years in the intervention group and 38.0 ± 16.4 years in the control group. The mean time since SCI in the intervention group was 5.4 ± 2.1 months, and 5.0 ± 2.1 months in the control group. The mean upper extremity motor score was 31.1 ± 12.0 years in the intervention group, and 31.6 ± 12.8 years in the control group. At baseline GRASSP, BBT, SCIM-III, and different types of MVC were not significantly different between the control and intervention groups (*p* > 0.05 for any comparisons, Table 2). The data were given in Table 3 for each group.

### 3.2. Functionality Assessment of Upper Extremity

The baseline total GRASSP score was not significantly different between the intervention and the control group (Table 2), and it showed significant improvement in both groups post-intervention, and the improvement was maintained during follow-up (*p* < 0.05, Figure 2A). However, the change score showed that improvements were significantly higher in the intervention than control group post-intervention and follow-up (*p* < 0.05, Figure 2B).

GRASSP Subtests. The upper extremity strength, prehension ability, and prehension performance improved significantly in both groups (*p* < 0.05 for all the comparisons, Table 3) and this improvement was maintained during follow-up (*p* < 0.05, Table 3). In the case of the change score comparison in the strength and prehension ability subtest, the improvement was significantly higher in the intervention than in the control group (*p* < 0.05; Figure 3A and Figure 3B, respectively).

Box and Block test. There were 13 subjects in the intervention group and 12 subjects in the control group who had the capacity to manipulate with both hands. There were significant improvements in both groups after the intervention, which were maintained during follow-up. Change scores were not significantly different between groups at any time of evaluation (*p* > 0.05, Table 3). 

SCIM III. There were significant improvements in both groups after intervention and were maintained during follow-up (*p* < 0.05). Changes score was not significant between groups at any time (*p* > 0.05, Table 3).

### 3.3. Maximal Voluntary Contraction (MVC)

Ten and nine subjects in the intervention group had the capacity, at baseline assessment, to perform the three grip patterns with right and left hands, respectively. In the control group, there were nine and ten subjects, respectively. Both groups improved significantly in cylindrical grip at post-intervention and this improvement was maintained during follow-up (*p* < 0.05, Table 3). In the case of lateral pinch force, only the intervention group improved significantly at post-intervention and follow-up (Figure 4A), whereas tip-to-tip pinch force was significantly different only in the follow-up (Figure 4B). The change score was not significantly different between groups in any grip pattern and at any time point evaluation (*p* > 0.05, Table 3). 

### 3.4. Neurological Assessment

The UEMS and total motor score was improved significantly in both groups post and during follow-up (*p* < 0.05, Table 3). Changes score was not significant between both groups post or during follow-up evaluation (*p* > 0.05, Table 3). The sensitive score did not change significantly in any group (*p* > 0.05, Table 3).

### 3.5. Quality of Life Assessment

There were no significant improvements in any quality of life dominion at any time of evaluation (*p* > 0.05, Table 3). 

### 3.6. Adverse Effects

All subjects described the tSCS as a continuous tingling sensation in the neck and arms. With tSCS, 2 of 15 subjects reported worsening of spasticity in upper limbs during stimulation; one subject reported feeling nauseous; two coughed and the other one reported an increased tingling sensation in the right lower limb. We had to reduce the stimulation intensity of tSCS in three subjects because of disturbing sensation in the stimulated area and in the other one; the stimulation intensity was reduced not to induce cough. Mild redness of skin was observed in all the subjects under the stimulation electrodes on the neck, which resolved within 5 min after the end of tSCS. Two subjects suffered dysreflexia during tSCS and we reduced the tSCS intensity to avoid it. In a few days, the blood pressure was regulated with reduced tSCS intensity. 

## 4. Discussion

According to our knowledge, this is the first study in the literature where tSCS was combined with Armeo^®^Power (intervention group), and the results were compared with the data from the control group (Armeo^®^Power alone). Until now only Inanici et al. [26] studied the effect of hand training with and without tSCS in 6 subjects with chronic cSCI. They reported significant improvements in GRASPP strength; GRASSP prehension ability and lateral pinch force with 24 sessions of tSCS combined with hand training in comparison to hand training alone [26]. 

Our study is the largest number of patients studied using tSCS compared to a control group for the upper extremities. Most of the studies of tSCS at the cervical spinal cord combined with hand training were case studies [25,27], clinical trials without a control group [23], or cross-over studies [26]. All of them have sample sizes of a maximum of six subjects with more than one year since cervical SCI. Our study is a randomized mix of parallel-group and a crossover clinical trial with a big sample of subjects with cervical spinal cord injury (15 in tSCS and 14 in the control group) with time since cSCI being less than 1 year of evolution. 

We stimulated the cervical spinal cord at two segments of the cervical spinal cord (at C3-C4 and C6-C7) as described by previous studies [23,25,26]. The other studies used just one segment of stimulation at C7-T1 [27], at C5-C6 [54], at C5-T2 [25], or at C5 [24]. In spite of one or two segments of cervical cord stimulation and/or at any level from C3 to T1, all studies reported significant improvement in the upper limb. 

In our study, the stimulation waveform was biphasic, rectangular, at a frequency of 30 Hz, similar to other studies that used cervical tSCS [23,25,26,54]. The carrier frequency used was 10 KHz as in the other studies [23,25,26,27], except in Benavides et al. [54], where 5 KHz was used, and in some studies used without any carrier frequency [24]. 

Using tSCS, the previous studies reported a significant improvement in hand grip and lateral pinch force [23,24,25,26], AIS motor and sensorial score [23,26], GRASSP and specifically in strength, prehension ability and performance subtests [25,26,27], self-care item of the SCIM III [25,26], and in the quality of life [25,26]. Our SCI subjects had been injured for less than one year, which could explain the significant improvement in the control group. The improvement, however, was significantly higher for the intervention than for the control group. It was previously published that using tSCS combined with activity-based rehabilitation, functional changes emerge more rapidly and to a greater degree than in isolation because the activity may help to enhance neuroplasticity [26,45]. In these studies, the hand training consisted in stretching, active-assisted exercises, fine and gross motor skills [25,26,27], MVC handgrip [23], isolated finger movements, and bimanual task performance [26]. The number of sessions among studies has differed from eight sessions to 45, with 1–2 h per session. In our study for hand training, we used upper extremity robotic therapy Armeo^®^Power which offers a more structured and homogenous intervention, with more repetition of specific tasks than activity-based rehabilitation. Furthermore, the virtual reality aspect favors the motivation of subjects to endure a greater number of repetitions during training.

Our results confirm the hypothesis that tSCS can modulate, non-invasively, cervical spinal networks and that possibly allows greater access to supraspinal control to cervical somatosensory networks shown in SCI [54] and healthy subjects [45]. We suggest that two of the main mechanistic events in neuromodulation are, first to acutely elevate the excitability and plasticity of residual neural networks of spinal and supraspinal neuronal networks and/or through the recruitment of afferent fibers from the posterior roots of the cord [37,55], and secondly, to chronically shape with training to the more plastic network connectivity toward a more normal coordinated functional state guided via the use-dependent mechanism. These activations cause interneurons and motor neurons to approach their activation threshold [35,56]. Motor neurons that are close to the threshold are easier to activate via intact but previously inactive residual descending pathways from the brain and proprioceptive input from peripheral afferents projecting to spinal networks [26]. This makes these neural structures more likely to respond to the SCI-limited downward drive and improve supraspinal control. 

We did not find any changes in the independence of basic activities of daily living and quality of life, possibly because of the short duration of the study (2 weeks). The aim of the study was to find changes in motor functions in the upper extremities, and we also maintained the duration shorter because the study was carried out during the COVID-19 pandemic, making it necessary to minimize long physical contact. This situation contributed to the loss of data during the follow-up period, reflecting the level of retention of clinical improvements maintained during the follow-up of 2 weeks. In other studies, the maintained improvements in upper extremities were maintained for a long term, until 3 or 6 months [26,27,28].

There are several limitations of this study: (i) We had limited staff numbers to realize the study was “blinding” due to the COVID-19 pandemic, and the blindness of SCI subjects was not performed since they were aware stimulation was on during the EMG recordings, as they were receiving the stimulation. (ii) Eight subjects were crossover: five from the control to the intervention and three from the intervention to the control group. We considered those subjects as new candidates because of a least or more than 4 weeks of washout period between each experiment. (iii) The study was conducted in an SCI population at a subacute stage, so there is a chance for neurological improvement. Both groups were homogenous in terms of severity, motor strength, and time since injury in order to reduce variability.

In conclusion, our results indicate that cervical tSCS, combined with upper robotic therapy, facilitated higher functional levels of motor tasks and can be beneficial for functional and motor recovery in spinal cord injury subjects. At a more conceptual level, our findings show the potential advantages of combining rehabilitation strategies with non-invasive spinal cord stimulation techniques to optimize outcomes. These beneficial effects were achieved with good tolerability. Challenges that remain include exploring the optimal dose and spinal cord segments for stimulation, the timing of stimulation, and developing better strategies for tSCS, according to the characteristics of the subject. For this, more studies are needed to improve the implementation of tSCS, study randomization, and translation to the clinical setting in the big number of SCI subjects. 

## Figures and Tables

**Figure 1 biomedicines-11-00589-f001:**
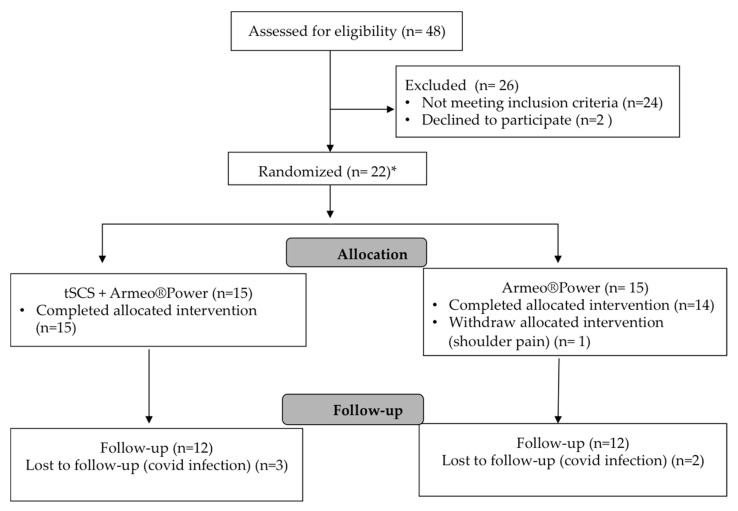
Flow diagram showing the number of subjects involved in each phase of the study. * 8 subjects were studied in both experiments (crossover) after at least 4 weeks of washout period. * crossover: five from the control group to the intervention group, and three from the intervention group to the control group.

**Figure 2 biomedicines-11-00589-f002:**
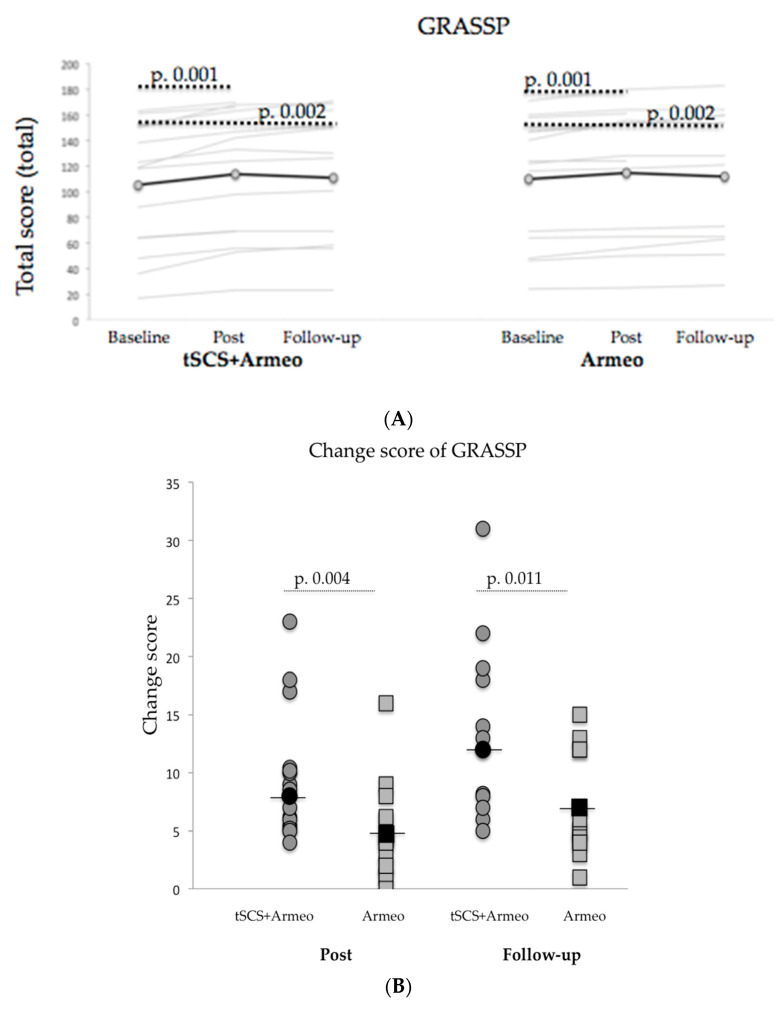
(**A**): GRASSP scores. Thin grey lines represent each subject and a thick black line is the mean of the group. The *p*-value according to the Wilcoxon test; (**B**): Changes scores of GRASSP, and *p*-value according to Mann-Whitney U test. The grey circles represent the change score of each subject in the intervention group and squares in the control group. Black circles and squares are the mean of the group.

**Figure 3 biomedicines-11-00589-f003:**
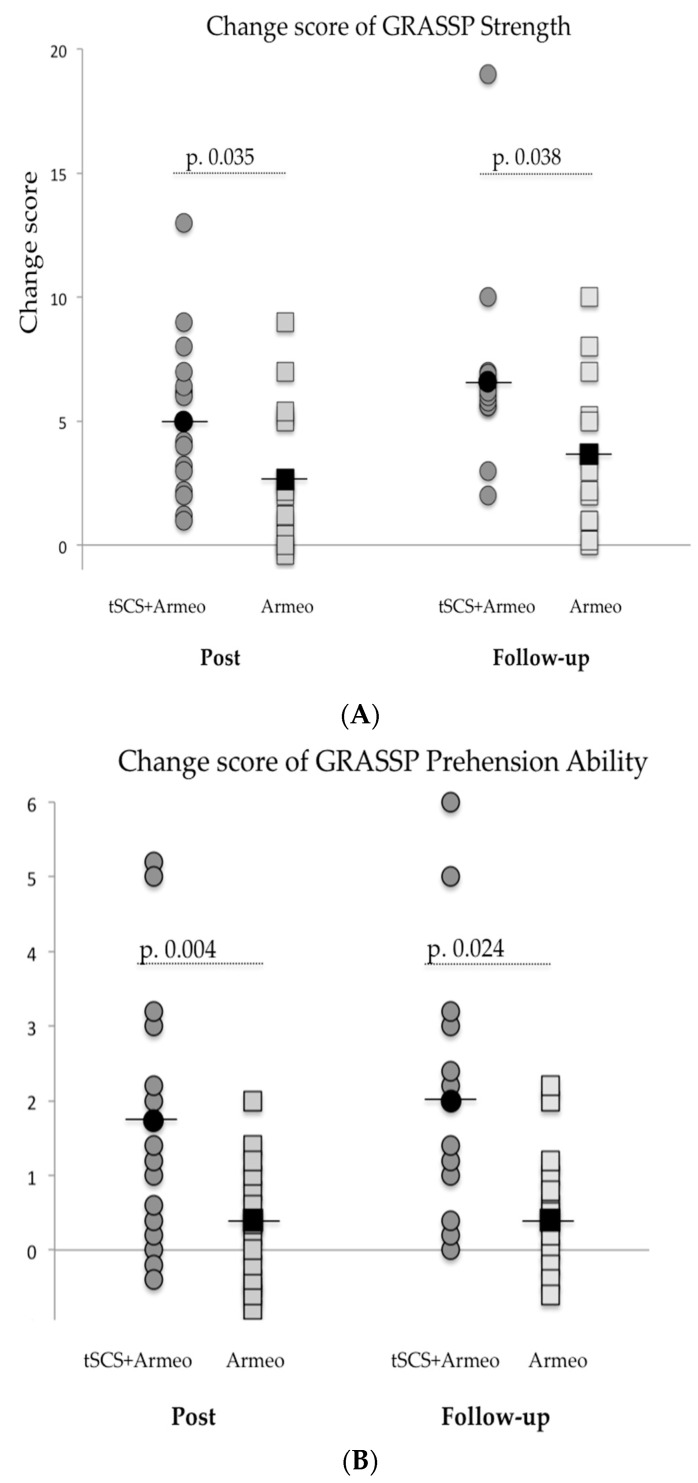
(**A**): Changes scores of strength subtest GRASSP; (**B**): Changes scores of prehension ability subtest GRASSP. Grey circles and squares represent each subject. Black circles and squares are the mean of the group. The *p*-value according to the Mann-Whitney U test.

**Figure 4 biomedicines-11-00589-f004:**
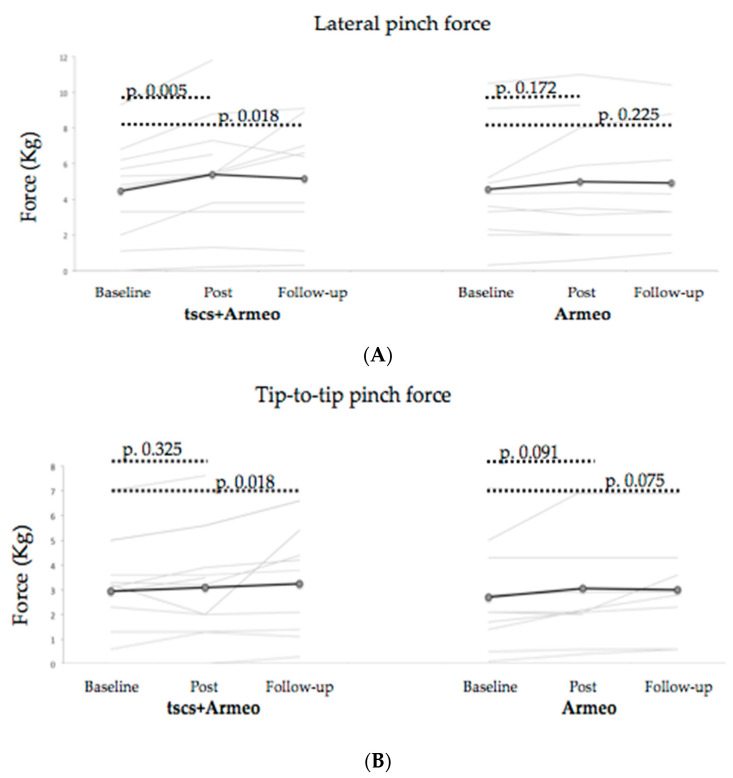
(**A**): Lateral pinch force score. The *p*-value according to the Wilcoxon test; (**B**): Tip-to-tip pinch force. The *p*-value according to the Wilcoxon test.

**Table 1 biomedicines-11-00589-t001:** Subject’s demographics and clinical characteristics and intensity of stimulation at C3-C4 and C6-C7 vertebral segments. The last two columns consist of the intensity of stimulation of each of the segment levels that received electrical stimulation.

Subject	Age	Gender	NLI	AIS	Total UEMS	Etiology	Time since SCI (Months)	Washout (Days)	Intensity tSCS at C3-C4 (mA)	Intensity tSCS at C6-C7 (mA)
					Intervention group (tSCS + Armeo^®^Power)					
1 **	46	M	C4	C	15	Trauma	4	59	80	80
2 **	25	M	C4	A	11	Trauma	4	111	67	86
4 *	36	M	C5	B	14	Trauma	6		74	86
6	36	M	C7	D	41	Trauma	6		63	85
8	28	M	C5	A	28	Trauma	4		86	86
10	28	M	C5	C	26	Trauma	6		63	85
12 *	38	M	C4	B	48	Trauma	5		54	77
15 *	56	M	C4	C	33	Medical	9		86	86
17 *	21	M	C5	D	42	Trauma	5		39	45
19	60	M	C5	D	43	Trauma	3		86	86
20 **	21	M	C6	B	22	Trauma	8	42	61	72
26 *	22	F	C7	C	43	Trauma	5		85	86
27	55	M	C3	D	41	Trauma	4		86	86
28	47	M	C6	D	31	Trauma	10		86	86
29	42	M	C4	D	31	Trauma	3		80	76
					Control group (Armeo^®^Power)					
3 *	36	M	C4	B	12	Trauma	5	30		
5 **	46	M	C4	C	17	Trauma	6			
7 **	25	M	C4	A	15	Trauma	8			
9	36	F	C4	C	37	Trauma	6			
11 *	38	M	C7	B	48	Trauma	3	58		
13 *	56	M	C4	C	32	Medical	6	79		
14 *	21	M	C5	D	42	Trauma	4	30		
16	70	M	C5	C	29	Trauma	8			
18	58	M	C6	A	44	Trauma	6			
21 **	21	M	C6	B	22	Trauma	9			
22	53	M	T1	B	47	Medical	3			
23	32	M	C5	D	40	Trauma	3			
24	18	M	C6	B	17	Trauma	4			
25 *	22	F	C7	C	41	Trauma	4	35		
*p*-value	0.43	0.19	0.63	0.61	0.51	0.19	0.89			

*, ** crossover: * from control group to intervention group, **: from intervention group to control group; M: male; F: female; NLI: neurologic level of injury; UEMS: upper extremity motor score. Duration of washout from first to second experimental group.

**Table 2 biomedicines-11-00589-t002:** The *p*-value of different functional assessments between both groups pre-intervention condition.

Outcome Measure	Upper Extremity	*p*-Value
GRASSP	R	0.631
	L	0.827
	R + L	0.861
BBT	R	0.740
	L	0.615
	R + L	0.740
SCIM III	R	-
	L	-
	R + L	0.194
MVC: cylindrical grip force	R	0.772
	L	0.982
	R + L	0.757
MVC: lateral pinch force	R	0.876
	L	0.705
	R + L	0.540
MVC: tip-to-tip pinch force	R	0.559
	L	0.947
	R + L	0.740

R: right upper extremity; L: left upper extremity; R + L: sum of right and left upper extremities.

**Table 3 biomedicines-11-00589-t003:** Data and statistics of all the measures for intervention and control group.

Outcome Measures	Intervention Group (tSCS + Armeo^®^Power)			Control Group (Armeo^®^Power)		Change Score	
	Time	Mean ± SD	*p* Value	Mean ± SD	*p* Value	*p* Value Baseline-Post between Intervention vs. Control Group	*p* Value Baseline-Follow between Intervention vs. Control Group
GRASSP strength (R)	Baseline Post Follow-up	26.13 ± 12.95 28.13 ± 12.90 26.75 ± 12.76	**0.001 ^$^** **0.002 ^$^**	28.64 ± 14.77 29.57 ± 15.11 27.00 ± 14.43	**0.016 ^$^** **0.011 ^$^**	**0.022**	**0.024**
GRASPP strength (L)	Baseline Post Follow-up	28.53 ± 13.33 31.53 ± 13.29 31.33 ± 13.94	**0.001 ^$^** **0.002 ^$^**	28.07 ± 12.12 29.78 ± 13.32 30.08 ± 14.33	**0.007 ^$^** **0.005 ^$^**	**0.110**	**0.073**
GRASSP strength (R+ L)	Baseline Post Follow-up	54.67 ± 24.31 59.67 ± 24.59 58.08 ± 25.26	**0.001 ^$^** **0.002 ^$^**	56.71 ± 25.83 59.36 ± 27.24 57.08 ± 28.33	**0.007 ^$^** **0.005 ^$^**	**0.035**	**0.038**
GRASSP sensation (R)	Baseline Post Follow-up	9.53 ± 3.89 10.33 ± 3.37 10.25 ± 3.52	**0.039 ^$^** 0.066 ^$^	10.21 ± 3.53 10.00 ± 3.57 9.83 ± 3.59	0.607 *	**0.014**	0.095
GRASSP sensation (L)	Baseline Post Follow-up	9.93 ± 3.49 10.27 ± 3.39 10.33 ± 3.68	0.202 *	10.71 ± 3.20 10.71 ± 3.20 10.50 ± 3.42	1.000 *	0.233	0.166
GRASSP Sensation (R + L)	Baseline Post Follow-up	19.47 ± 7.26 20.60 ± 6.74 20.58 ± 7.14	**0.017 ^$^** **0.042 ^$^**	20.93 ± 6.53 20.71 ± 6.56 20.33 ± 6.88	0.717 *	**0.011**	0.113
GRASSP prehension ability: cylindrical (R)	Baseline Post Follow-up	1.67 ± 1.67 1.73 ± 1.71 1.75 ± 1.71	0.156 *	2.00 ± 1.71 1.93 ± 1.73 1.58 ± 1.62	0.368 *	0.164	0.088
GRASSP prehension ability cylindrical (L)	Baseline Post Follow-up	2.00 ± 1.73 2.13 ± 1.77 2.17 ± 1.80	0.135 *	1.93 ± 1.68 2.07 ± 1.77 2.25 ± 1.81	0.135 *	0.942	1.000
GRASSP prehension ability: cylindrical grasp (R + L)	Baseline Post Follow-up	3.67 ± 3.22 3.87 ± 3.27 3.92 ± 3.42	0.082 *	3.93 ± 3.07 4.00 ± 3.09 3.83 ± 3.30	0.368 *	0.563	0.248
GRASSP prehension ability: lateral key pinch (R)	Baseline Post Follow-up	1.53 ± 1.24 2.00 ± 1.56 1.67 ± 1.37	**0.024 ^$^** **0.046 ^$^**	1.86 ± 1.51 1.86 ± 1.51 1.67 ± 1.37	1.000 ^$^ 0.317 ^$^	**0.009**	0.140
GRASSP prehension ability: lateral key pinch (L)	Baseline Post Follow-up	2.00 ± 1.56 2.40 ± 1.64 2.25 ± 1.60	**0.041 *^,^** ** ^⌘^ **	2.21 ± 1.31 2.43 ± 1.50 2. 33 ± 1.57	0.083 0.157	0.620	0.286
GRASSP prehension ability: lateral key pinch (R + L)	Baseline Post Follow-up	3.53 ± 2.69 4.40 ± 3.11 3.91 ± 2.87	**0.017 ^$^** **0.026 ^$^**	4.07 ± 2.70 4.28 ± 2.87 4.00 ± 2.79	0.097 *	0.092	0.131
GRASSP prehension ability: tip-to-tip pinch (R)	Baseline Post Follow-up	1.73 ± 1.33 1.93 ± 1.39 1.58 ± 1.16	0.247 *	1.78 ± 1.31 1.78 ± 1.31 1.66 ± 1.30	0.368 *	0.163	0.596
GRASSP prehension ability: tip-to-tip pinch (L)	Baseline Post Follow-up	1.93 ± 1.33 2.40 ± 1.50 2.33 ± 1.56	**0.020 ^$^** **0.034 ^$^**	1.93 ± 1.33 2.00 ± 1.36 2.08 ± 1.56	**0.040 ^$^** 0.167 ^$^	**0.040**	0.167
GRASSP prehension ability: tip-to-tip pinch (R + L)	Baseline Post Follow-up	3.67 ± 2.44 4.33 ± 2.66 3.91 ± 2.43	**0.016 ^$^** **0.026 ^$^**	3.71 ± 2.40 3.78 ± 2.42 3.75 ± 2.70	**0.017 ^$^** 0.111 ^$^	**0.017**	0.111
GRASSP total prehension ability (R)	Baseline Post Follow-up	4.93 ± 4.08 5.67 ± 4.50 5.00 ± 3.95	**0.010 ^$^** **0.016 ^$^**	5.64 ± 4.36 5.57 ± 4.36 4.92 ± 4.16	0.368 *	**0.001**	**0.021**
GRASSP total prehension ability (L)	Baseline Post Follow-up	5.93 ± 4.33 6.93 ± 4.62 6.75 ± 4.71	**0.011 ^$^** **0.016 ^$^**	6.07 ± 4.14 6.50 ± 4.41 6.67 ± 4.81	**0.039 *^,^** ** ^⌘^ **	0.140	0.084
GRASSP total prehension ability (R + L)	Baseline Post Follow-up	10.87 ± 7.98 12.60 ± 8.74 11.75 ± 8.38	**0.002 ^$^** **0.008 ^$^**	11.71 ± 7.95 12.07 ± 8.16 11.58 ± 8.64	0.059 ^$^ **0.038 ^$^**	**0.004**	**0.024**
GRASSP prehension Performance (R)	Baseline Post Follow-up	10.07 ± 6.45 10.87 ± 6.43 11.25 ± 6.94	0.066 ^$^ **0.026 ^$^**	10.50 ± 6.78 11.21 ± 6.74 10.25 ± 6.56	0.059 ^$^ **0.038 ^$^**	0.887	0.077
GRASSP prehension performance (L)	Baseline Post Follow-up	11.20 ± 7.51 12.07 ± 7.25 12.42 ± 7.21	**0.025 ^$^** **0.011 ^$^**	10.07 ± 6.85 11.28 ± 7.03 12.50 ± 6.93	**0.011 ^$^** **0.008 ^$^**	0.890	0.638
GRASSP prehension performance (R + L)	Baseline Post Follow-up	21.27± 13.44 22.93 ± 13.22 23.67 ± 18.84	**0.019 ^$^** **0.004 ^$^**	20.57 ± 12.69 29.94 ± 12.57 22.75 ± 13.14	**0.004 ^$^** **0.005 ^$^**	0.639	0.448
GRASSP (R)	Baseline Post Follow-up	50.67 ± 25.08 55.00 ± 25.19 53.25 ± 25.10	**0.001 ^$^** **0.016 ^$^**	55.00 ± 27.67 56.36 ± 27.71 52.00 ± 26.40	**0.016 ^$^** **0.015 ^$^**	**0.003**	**0.003**
GRASSP (L)	Baseline Post Follow-up	55.60 ± 26.65 60.80 ± 26.67 60.83 ± 27.67	**0.001 ^$^** **0.002 ^$^**	54.93 ± 24.12 58.28 ± 25.81 59.75 ± 27.37	**0.002 ^$^** **0.002 ^$^**	0.108	0.308
GRASSP (R + L)	Baseline Post Follow-up	106.27 ± 49.92 115.80 ± 49.91 114.08 ± 51.25	**0.001 ^$^** **0.002 ^$^**	109.93 ± 49.54 114.64 ± 51.05 111.75 ± 52.94	**0.001 ^$^** **0.002 ^$^**	**0.004**	**0.011**
BBT (R)	Baseline Post Follow-up	35.31 ± 16.96 40.23 ± 15.66 43.10 ± 15.95	**0.004 ^&^** **0.000 ^&^**	28.14 ± 19.47 33.00 ± 21.98 31.00 ± 21.48	**0.008 ^&^** **0.007 ^&^**	0.715	0.164
BBT (L)	Baseline Post Follow-up	36.54 ± 18.18 40.00 ± 17.43 45.20 ± 16.18	**0.008 ^&^** **0.000 ^&^**	27.71 ± 20.60 31.36 ± 21.74 34.58 ± 20.79	**0.010 ^&^** **0.003 ^&^**	0.826	0.163
BBT (R+ L)	Baseline Post Follow-up	66.71 ± 37.67 74.50 ± 36.96 80.27 ± 39.00	**0.002 ^&^** **0.000 ^&^**	55.86 ± 38.15 64.36 ± 41.72 65.58 ± 41.41	**0.003 ^&^** **0.001 ^&^**	0.629	0.185
MVC: cylindrical grasp force (R)	Baseline Post Follow-up	12.51 ± 10.96 14.37 ± 10.91 11.84 ± 7.95	**0.007 ^$^** **0.017 ^$^**	11.37 ± 9.13 12.23 ± 9.91 10.87 ± 6.37	**0.046 ^$^** **0.018 ^$^**	0.152	0.247
MVC: cylindrical grasp force (L)	Baseline Post Follow-up	9.54 ± 5.50 10.90 ± 5.81 12.77 ± 7.31	**0.009 ^$^** **0.008 ^$^**	9.40 ± 6.26 10.92 ± 7.35 12.49 ± 6.65	0.059 ^$^ **0.036 ^$^**	0.860	0.211
MVC: cylindrical grasp force (R + L)	Baseline Post Follow-up	20.90 ± 12.25 23.96 ± 12.64 23.29 ± 11.21	**0.005 ^$^** **0.008 ^$^**	19.63 ± 12.22 21.93 ± 14.16 22.00 ± 12.64	**0.037 ^$^** **0.017 ^$^**	0.398	0.083
MVC: lateral pinch force (R)	Baseline Post Follow-up	2.67 ± 2.41 3.11 ± 2.60 2.26 ± 1.54	**0.012 ^$^** 0.093 ^$^	2.39 ± 2.37 2.82 ± 2.51 2.28 ± 1.88	**0.042 ^$^** 0.114 ^$^	0.837	0.815
MVC: lateral pinch force (L)	Baseline Post Follow-up	2.24 ± 1.46 2.82 ± 1.44 3.55 ± 1.92	**0.008 ^$^** **0.028 ^$^**	2.40 ± 1.45 2.44 ± 1.64 2.91 ± 1.54	0.368 *	**0.033**	0.090
MVC: lateral pinch force (R + L)	Baseline Post Follow-up	4.46 ± 2.70 5.39 ± 3.29 5.17 ± 3.29	**0.005 ^$^** **0.018 ^$^**	4.55 ± 3.14 4.98 ± 3.46 4.91 ± 3.30	0.341 *	0.138	0.107
MVC: tip-to-tip pinch force (R)	Baseline Post Follow-up	1.64 ± 1.60 1.72 ± 1.69 1.57 ± 1.46	0.232 ^$^ **0.031 ^$^**	1.31 ± 1.36 1.70 ± 1.84 1.30± 1.33	0.128 *	0.435	0.815
MVC: tip-to-tip pinch force (L)	Baseline Post Follow-up	1.59± 0.89 1.68 ± 0.91 2.09 ± 1.11	0.360 ^$^ **0.021 ^$^**	1.53 ± 1.19 1.52 ± 0.91 1.86 ± 1.00	0.125 *	0.647	0.342
MVC: tip-to-tip pinch force (R + L)	Baseline Post Follow-up	2.94 ± 1.96 3.09 ± 2.15 3.25 ± 2.13	0.325 ^$^ **0.018 ^$^**	2.71 ± 2.17 3.05 ± 2.32 3.00 ± 2.05	**0.034 *^,^** ** ^⌘^ **	0.776	0.358
Upper motor extremity AIS	Baseline Post Follow-up	31.27 ± 11.83 33.20 ± 11.58 33.00 ± 12.99	**0.001 ^&^** **0.002 ^&^**	31.64 ± 12.83 32.86 ± 13.17 32.17 ± 13.74	**0.009 ^&^** **0.002 ^&^**	0.241	0.129
Total Motor Score AIS	Baseline Post Follow-up	50.27 ± 27.44 53.33 ± 28.43 50.25 ± 27.45	**0.017 ^&^** **0.015 ^&^**	45.50 ± 26.52 47.71 ± 27.50 42.33 ± 24.85	0.051 ^&^ **0.023 ^&^**	0.583	0.394
Total Sensitive Score AIS	Baseline Post Follow-up	68.00 ± 24.37 67.67 ± 25.05 66.92 ± 23.80	0.628 ^&^ 0.199 ^&^	64.67 ± 28.31 64.33 ± 27.72 63.17 ± 25.96	0.547 ^&^ 0.191 ^&^	0.456	0.759
SCIM III	Baseline Post Follow-up	53.87 ± 26.65 57.40 ± 27.57 52.75 ± 26.72	**0.007 ^&^** **0.033 ^&^**	41.28 ± 24.21 43.00 ± 24.53 43.00 ± 24.53	**0.019 ^&^** **0.019 ^&^**	0.208	0.130
WHO-QoL-BREF Quality of life	Baseline Post Follow-up	2.80 ± 1.15 2.86 ± 0.99 2.83 ± 1.11	0.449 *	2.38 ± 1.04 2.85 ± 1.14 2.54 ± 1,51	0.143 *	0.083	0.657
WHO-QoL-BREF Health satisfaction	Baseline Post Follow-up	3.00 ± 1.00 3.00 ± 1.00 2.83 ± 1.27	0.584 *	2.31 ± 1,18 2.69 ± 1.25 2.63 ± 1.43	0.116 *	0.067	0.101
WHO-QoL-BREF Physical Health	Baseline Post Follow-up	46.73 ± 14.07 49.33 ± 13.79 45.92 ± 15.86	0.267 *	41.61 ± 10.96 45.38 ± 13.28 42.18 ± 20.42	0.317 *	0.599	0.706
WHO-QoL-BREF Psychological	Baseline Post Follow-up	54.60 ± 19.74 54.60 ± 20.12 51.08 ± 20.43	0.670 *	52.00 ± 17.10 56.31 ± 17.85 50.55 ± 25.97	0.183 *	0.168	0.611
WHO-QoL-BREF Social relationships	Baseline Post Follow-up	53.73 ± 19.89 53.33 ± 21.15 46.42 ± 22.50	0.101 *	53.92 ± 18.74 15. 12 ± 51.85 48.27 ± 22.40	0.444 *	0.289	0.972
WHO-QoL-BREF Environment	Baseline Post Follow-up	48.40 ± 18.53 48.73 ± 16.50 48.42 ± 15.49	0.453 *	47.23 ± 11.36 46.15 ± 15.25 46.15 ± 15.25	0.867 *	0.745	0.562

*: The *p*-value according to the Friedmann test; ^$^: Wilcoxon-*t*-test; ^&^: T-test; **^⌘^**: Wilcoxon was 0.066 between baseline vs. post-intervention and vs. follow-up after the significant Friedman test. For all changes scores comparison, *p*-value according to Mann-Whitney-U test.

## Data Availability

The data presented in this study are available on request from the corresponding author.

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
