# Peer review of "Transcutaneous Cervical Spinal Cord Stimulation Combined with Robotic Exoskeleton Rehabilitation for the Upper Limbs in Subjects with Cervical SCI: Clinical Trial"

_biomedicines, 2023, doi:10.3390/biomedicines11020589_

Round 1
Reviewer 1 Report
This manuscript describes a study of Armeo hand therapy with or without cervical spinal cord stimulation in a group of people with cervical SCI. It is described as a randomized controlled trial. It has a mix of parallel-group and crossover participants.
Multiple issues listed below:
The extremely major issue with this paper is that it is a mix of parallel group and crossover, yet it really does not make that clear, nor does it account for that at all in the statistical analysis. In fact, the words “crossover” and “washout” are not mentioned at all in this manuscript!! Nor is this depicted in Figure 1 CONSORT diagram. Unfortunately, that is not acceptable.
The statistical analysis does not account at all for the fact that 8 participants did a crossover whereas the other 21 did one or the other treatment. Nor is anything mentioned about whether the study was meant to be a crossover a priori, or whether some participants were invited back to do a second intervention after completing the first, etc.
Crossover studies are difficult to analyze. There is almost always missing data, as in this study where some participants did a crossover and others didn’t. Usually, more sophisticated analysis for example using a general linear model is required. There is supposed to be an analysis of washout/carryover effects and order effects. None of this was done here.
Furthermore, what does ‘at least 4 weeks’ washout mean? This is a subacute SCI population. Given that there is often ongoing improvement during the subacute period, performing a crossover study with ‘at least’ 4 weeks washout raises huge issues regarding order effect etc.
This manuscript lacks a Limitations section in its Discussion. All scientific manuscripts should include a Limitations section before undergoing peer review.
Please provide an analysis of just the 21 participants who underwent one intervention.
The introduction and the methodology are otherwise clearly written – except the major issues with trial design and analysis.
Please explain the biological reason why left and right scores should be analyzed separately in a group of SCI participants undergoing bilateral upepr extremity training? Was this decision made post hoc? Then that should be stated.
I did not understand the statement, “The clinical assessments were performed on different days over the course of a week to avoid fatigue.” ???
Author Response
Point 1:The extremely major issue with this paper is that it is a mix of parallel group and crossover, yet it really does not make that clear, nor does it account for that at all in the statistical analysis. In fact, the words “crossover” and “washout” are not mentioned at all in this manuscript!! Nor is this depicted in Figure 1 CONSORT diagram. Unfortunately, that is not acceptable.
Response 1: Now we used the terminology of “crossover” and “washout” which before were described by other terminology in whole text "8 subjects were studied in both interventions " in the table 1 and figure 1 and "at least 4 weeks of break between both intervention" in the methodology.
Point 2:The statistical analysis does not account at all for the fact that 8 participants did a crossover whereas the other 21 did one or the other treatment. Nor is anything mentioned about whether the study was meant to be a crossover a priori, or whether some participants were invited back to do a second intervention after completing the first, etc.
Response 2: We added more information clarifying this point:"The experimental design, at the beginning was a randomized, controlled study, which consisted of two groups: (i) intervention group: tSCS combined with robotic exoskeleton (Armeo®Power)and (ii) control group: robotic exoskeleton (Armeo®Power) alone. We used a computer-generated list as randomization strategy. Assignment of the subjects to the treatment interventions was random. But subjects were given the possibility to participate in both group."
Point 3:Crossover studies are difficult to analyse. There is almost always missing data, as in this study where some participants did a crossover and others didn’t. Usually, more sophisticated analysis for example using a general linear model is required. There is supposed to be an analysis of washout/carryover effects and order effects. None of this was done here.
Response 3: Eight subjects who received the 2 interventions, since they had at least four or more than 4 weeks of washout period, we considered that they could be considered as different subjects for the next experimental group. For this reason the most direct way was to compare the 2 groups (tSCS+ARMEO vs. ARMEO).
Point 4:Furthermore, what does ‘at least 4 weeks’ washout mean? This is a subacute SCI population. Given that there is often ongoing improvement during the subacute period, performing a crossover study with ‘at least’ 4 weeks washout raises huge issues regarding order effect etc.
Response 4: Thank you for this point: Like we have explained in point 3,the eight subjects who did both interventions they can be considered as different subjects. And most of the subject crossover from control group means that less time since SCI could have more improvement but our result showed contrary they improved significantly less than the control group. This information added in the limitation of the study.
Point 5:This manuscript lacks a Limitations section in its Discussion. All scientific manuscripts should include a Limitations section before undergoing peer review.
Response 5: We added a limitation section: “They are several limitations of this study: (i) We had a limited staff numbers to realize the study "blinding"due to COVID pandemic and the blindness of SCI subjects was not performed since they were aware if stimulation was on because during EMG recordings, they received the stimulation. (ii) Eight subjects were crossover: five from control to intervention and three from intervention to control group. We considered that those subjects as a new candidate because of a least or more than 4 weeks of washout period between each experiment. (iii) The study was done in a SCI population at a subacute stage, so there is a chance for neurological improvement. Both groups were homogeneus in terms of severity, motor strength and time since injury in order to reduce variability.
Point 6:Please provide an analysis of just the 21 participants who underwent one intervention.
Response 6: We did an analysis of the 21 participantswithout crossover. But GRASSP total was almost significantly different between both groups (p.0.06). Because of this reason, we decided to comment the results of the groups without crossover here but not in the manuscript.
Result without crossover:
There was significant improvement in both groupsfor UEMS, total motor score, GRASSP total, strength, total prehension ability and performance, BBT, MVC: cylindrical grasp force, SCIM III.
In intervention group, also there were significant improvement for GRASSPsensation, lateral key pinch, cylindrical grasp, total prehension ability, MVC: lateral pinch force , MVC: tip to tip pinch force.
The change score was significantly higher in intervention group than control groupfor GRASSP total and GRASSP sensation.
Those results without crossover were similar to the results of crossover group with less point of significance in change scores.
Point 7:The introduction and the methodology are otherwise clearly written – except the major issues with trial design and analysis.
Response 7: We rewrote trial designas suggested, but we could not change the statistical analysis explained in the Point 3 and 6.
Point 8:Please explain the biological reason why left and right scores should be analysed separately in a group of SCI participants undergoing bilateral upper extremity training? Was this decision made post hoc? Then that should be stated.
Response 8: Limitations in upper extremity in cervical spinal cord injuries usually are different between left and right side. And in all tests, first value right and left in a separated way and then sum the results. Also training it´s not a bilateral task, since each upper extremity training 30 min separately.
Point 9:I did not understand the statement, “The clinical assessments were performed on different days over the course of a week to avoid fatigue.” ???
Response 9: The assessment in all the times (baseline, post-intervention and follow-up) has been performed during 3 days because there were a lot of tests (around 4-5 hours of duration) and the fatigue of the subject could affect the results.
The assessments were included:
- Upper extremity motor score (UEMS), AIS total motor score, AIS sensitive score took around 30-40 min
- Graded Redefined Assessment of Strength, Sensibility and Prehension (GRASSP, version 2) took around 30-40 min
- Box and Block test (BBT) took around 15 min
- Grip force by Maximal Voluntary Contraction (MVC) took around 20 min
- Spinal Cord Independence Measure III (SCIM III) took around 15 min
- Neurophysiological evaluation (threshold from APB for tSCS at two cervical segments)
We added this information in the text: "Total duration of clinical and neurophysiological assessments was around 4-5 hours. The clinical assessments were performed on 3 different days before the experiment to avoid fatigue."
Reviewer 2 Report
The authors successfully presented the results of the study on twenty-two subjects (patients?) who participated in the randomized, controlled intervention, consisting of an intervention group (tSCS combined with a robotic exoskeleton) and a control group (only robotic exoskeleton). They concluded that their findings indicate the potential advantages of combining rehabilitation strategies with non-invasive spinal stimulation techniques to optimize outcomes for the upper extremity.
The paper sounds, the results are convincing and presented with the knowledge of true scientific art.
The Introduction includes a perfect presentation of “the state of art”, especially lines 60-63 summarize carefully the contemporary discussed efficiency of tSCS. The hypotheses (lines 73-80 84-87) are convincing about the importance of the undertaken study.
M&M content enables the repetition of the study in another hospital or laboratory. Pity that the results were not statistically compared to the “real” control group of healthy volunteers. It would show the degree of the patient's impairment before and after the intervention.
Results - very well presented, very convincing, unfortunately the upper part of the Figure 2 with light lines is of the low quality
Discussion – very convincing, especially the neurophysiological explanation of tSCS mechanism of action in lines 398-406
Minor comments:
The abstract is sparing and does not contain data on the evaluation procedures used, which is missing. The phrase …“on motor strength and functionality”… is not informative enough, methods should be listed. The last keyword "functionality" may refer to everything, it doesn't fit the rest of the applied.
Introduction
Line 81 - …”Here, “… could be replaced with ..”The aim of study was a randomized,…” – the reader would be able easier to find the study's aim.
M&M
Line 94 …” (ii) a stable traumatic o no traumatic cervical SCI; “… or?
Line 97 …” with APB at least in one side”… shorts used the first time in the throughout text should be fully explained (not only here)
Line 99 …” (≥ 3 score on the Modified Ashworth scale (MAS)) - Modified Ashworth Scale – MAS – better?
Line 101 – peacemakers, electronic implants, episodes of the epilepsy not mentioned in exclusion criteria for tSCS therapy
Results
Line 269-270 … *: p value according to Friedmann test; $ : wilxocon-t-tes; ª: One-way ANOVA; t & : t-test; ⌘: wilcoxon was 0.066 between baseline vs post-intervention and vs. follow-up. p value according to Mann-Whitney-U test for all changes scores comparison”… - p-value, Big letters, small letters, names of tests…
Line 315 …” table 3).”… - Big letters, small letters…
Discussion
Lines 382-385 …” The results in previous studies showed a significant improvement in hand grip and lateral pinch force (25,25–27) ,AIS motor and sensorial score (25,27), GRASSP, specifically in strength, prehension ability and performance subtests (26–28) ,self-care item of the SCIM III (26,27) and in quality of life (26,27).”… - please rewrite…
Refs. are selected with great skill, but sometimes the authors use an editorial style not accepted by MDPI (sometimes they use journal abbreviations, sometimes they don't). Ref. 29 is incorrectly quoted.
Author Response
We appreciate very much the constructive comments and the correction of the reviewer.
Abstract
Point 1: The abstract is sparing and does not contain data on the evaluation procedures used, which is missing. The phrase …“on motor strength and functionality”… is not informative enough, methods should be listed. The last keyword "functionality" may refer to everything, it doesn't fit the rest of the applied.
Response 1:We modified this part and added more specific information about clinical outcome as suggested, but we had to maintain limit of word that was 200. “All subjects from each group were evaluated at baseline condition, after the last session, and then at two weeks after last session. We assessed graded redefined assessment of strength, sensibility and prehension(GRASSP), box & block test (BBT), spinal cord independence measure III (SCIM-III), maximal voluntary contraction (MVC), ASIA impairment scale (AIS) and WhoQol-Bref”;
Introduction
Point 2: Line 81 - …”Here, “… could be replaced with ..”The aim of study was a randomized,…” – the reader would be able easier to find the study's aim.
Response 2: it was done as suggested "The aimof the study was to determine the effects of cervical tSCS combining with robotic exoskeleton on motor strength and functionality of the upper extremity in subjects with cervical spinal cord injury".
Point 3: Line 94 …” (ii) a stable traumatic o no traumatic cervical SCI; “… or?
Response 3: It was corrected as suggested in the text: "it was traumatic or no traumatic cervical SCI" .
Point 4: Line 97 …” with APB at least in one side”… shorts used the first time in the throughout text should be fully explained (not only here)
Response 4: It was corrected as suggested.
Point 5: Line 99 …” (≥ 3 score on the Modified Ashworth scale (MAS)) - Modified Ashworth Scale – MAS – better?
Response 5: It was corrected in the manuscript.
Point 6: Line 101 – peacemakers, electronic implants, episodes of the epilepsy not mentioned in exclusion criteria for tSCS therapy
Response 6: They were added as suggested in the manuscript.
Results
Point 7: Line 269-270 … *: p value according to Friedmann test; $ : wilxocon-t-tes; ª: One-way ANOVA; t & : t-test; ⌘: wilcoxon was 0.066 between baseline vs post-intervention and vs. follow-up. p value according to Mann-Whitney-U test for all changes scores comparison”… - p-value, Big letters, small letters, names of tests…
Response 7: Those details were corrected as suggested in the manuscript.
Point 8: Line 315 …” table 3).”… - Big letters, small letters…
Response 8: Done as suggested.
Discussion
Point 9: Lines 382-385 …” The results in previous studies showed a significant improvement in hand grip and lateral pinch force (25,25–27) ,AIS motor and sensorial score (25,27), GRASSP, specifically in strength, prehension ability and performance subtests (26–28) ,self-care item of the SCIM III (26,27) and in quality of life (26,27).”… - please rewrite…
Response 9:Done as suggested: " The previous studies reported asignificant improvement in hand grip and lateral pinch force (27,27–29),AIS motor and sensorial score (27,29), GRASSP and specifically in strength, prehension ability and performance subtests (28–30), self-care item of the SCIM III (28,29)and in the quality of life (28,29).
Point 10: Refs. are selected with great skill, but sometimes the authors use an editorial style not accepted by MDPI (sometimes they use journal abbreviations, sometimes they don't). Ref. 29 is incorrectly quoted.
Response 10:Corrected in the Refs. as suggested.
Reviewer 3 Report
Reviewer Comments
Thank you very much for the opportunity to review the manuscript submission entitled: Transcutaneous cervical spinal cord stimulation combined with robotic exoskeleton rehabilitation for the upper limbs in subjects with cervical SCI. Clinical trial.
The aim of this study was to determine the effects of cervical tSCS when combined with robotic exoskeleton on motor strength and functionality of the upper extremity in subjects with cervical spinal cord injury. The data is interesting and it has a relevant rationale, however, some limitations and constructive comments are pointed below:
General comments
· Needs English editing
Specific comments
Title and Abstract
· Mention study design clearly – not just clinical trial.
· Include mean age of the participants in the abstract
· Clearly mention the clinical outcomes?
· Include significant values and the outcome values for between group comparisions.
· Include MeSH terms as keywords
Introduction
· Need more explanation on the scientific background and rationale for the investigation being reported.
· Include prespecified hypotheses of the study
Methods
· Present key elements of study design early in the paper.
· Describe the setting, locations, and relevant dates, including periods of recruitment,
· exposure, follow-up, and data collection
· Clearly define all outcomes, exposures, predictors, potential confounders, and effect modifiers. Give diagnostic criteria.
· Clearly explain about the randomization and Allocation concealment process.
· Why the design was did not considered subjects and investigators ‘blinding’ about treatment allocation. Explain? I did not see the explanation in the limitations section also.
· How well was the study done to minimize bias? Describe any efforts to address potential sources of bias
· Explain how the study size was arrived at?
Results
· The study should report any important differences in the composition of the study groups with regard to characteristics that could affect response to the intervention being investigated.
· Why did you not consider intention to treat analysis? Intention-to-treat analysis is a method for analyzing results in a prospective randomized study where all participants who are randomized are included in the statistical analysis and analyzed according to the group they were originally assigned, regardless of what treatment they received
Discussion:
· Discuss limitations of the study, taking into account sources of potential bias or imprecision. Discuss both direction and magnitude of any potential bias.
· Give a cautious overall interpretation of results considering objectives, limitations, multiplicity of analyses, results from similar studies, and other relevant evidence
· Discuss the generalizability (external validity) of the study results
Author Response
We appreciate very much the constructive comments and the correction of the reviewer.
Title and abstract
Point 1: Mention study design clearly – not just clinical trial.
Response 1: We decided not to change the title because of this study was randomized mixed parallel crossover clinical trial. This was explained in the manuscript now.
Point 2: Include mean age of the participants in the abstract
Response 2: The information was added in the abstract as suggested: “twenty-one subjects with cervical spinal cord injury (mean age 39 years) participated in the randomized, controlled intervention, consisting of an intervention group (n=15; tSCS+exoskeleton) and control group (n=14; exoskeleton)”.
Point 3: Clearly mention the clinical outcomes?
Response 3: Now, some clinical outcome information was given in the abstract, but we had to maintain limit of words maximum 200.
Point 4: Include significant values and the outcome values for between group comparisons.
Response 4: We added the significant values in groups and between groups but we could not give more detailed information because of limited words number (200words).
Point 5: Include MeSH terms as keywords
Response 5: We included MesH terms “upper extremity” and “robotics” as suggested
Introduction
Point 6: Need more explanation on the scientific background and rationale for the investigation being reported.
Response 6: We added more information on the scientific background and rationale for the investigation in the introduction.
Point 7: Include prespecified hypotheses of the study
Response 7:Now we rewrote this part: " Our hypothesis was that subjects who receive tSCS combined with rehabilitation assisted by upper limb robotic exoskeleton could improve motor strength and functionality more than those who received rehabilitation assisted with robotic exoskeleton alone."
Methods
Point 8: Present key elements of study design early in the paper.
Response 8: To study the motor and functional changes of upper limb in both groups, we used upper extremity motor score (UEMS) and AIS total score; for functional change: Graded Redefined Assessment of Strength, Sensibility and Prehension (GRASSP, version 2) which includes strength, sensation, prehension ability and prehension performance; manual dexterity measured by Box and Block test (BBT); Maximal Voluntary Contraction (MVC) and Spinal Cord Independence Measure III (SCIM III) (56)to evaluate an independence in activities of daily living
Point 9:Describe the setting, locations, and relevant dates, including periods of recruitment, exposure, follow-up, and data collection
Response 9: We added information about the details suggested by reviewer`; : ““The study was carried out in the Guttmann Institut during twenty-two months, starting on August 2020 and ending on March 2022. One or two subjects were recruited per month”. Data collection was performed for each patient after any assessments and data analysis of data was done after finishing the assessment of last patient's follow-up. We added this part in data and statistical analysis section.
Point 10:Clearly define all outcomes, exposures, predictors, potential confounders, and effect modifiers. Give diagnostic criteria.
Response 10: Due to the small size of the sample, cannot be studied how some potential confounders (severity and level of the lesion, age, gender and time since injury) can affect the outcomes. An attempt has been made to reduce this effect by homogenizing the groups (no significant differences in clinical and demographics between groups).
Pointe 11:Clearly explain about the randomization and Allocation concealment process.
Response 11:We used a computer-generated list as randomization strategy. But patients who wanted to participate in intervention or control group, they were permitted to realize both intervention. Assignment of the patients to the treatment interventions was random.
Point 12: Why the design was did not considered subjects and investigators ‘blinding’ about treatment allocation. Explain? I did not see the explanation in the limitations section also.
Response 12:“We had a limited staff numbers to realize "blinding" due to COVID pandemic and the blindness of SCI subjects was not performed since they were aware if stimulation was on becuse during EMG recordings, they received the stimulation".
Now, we added those details as limitations of the study.
Point 13: How well was the study done to minimize bias? Describe any efforts to address potential sources of bias
Response 13: To minimize bias, the study was done in homogeny groups for demographical data (age, sex, level, severity and duration of SCI, GRASSP, BBT, SCIM-III, different type of MVC). And a randomization allocation was used to distribute the subject to the experimental groups.
Point 14: Explain how the study size was arrived at?
Response 14:The aim was to study 15 cervical spinal cord injury subjects for each group. But duration of project was 2 years and we finalized it with difficulties due to COVID (locked-downs, withdrawal of patients).
Results
Point 15: The study should report any important differences in the composition of the study groups with regard to characteristics that could affect response to the intervention being investigated.
Response 15: There were no differences between both groups; the details are showed in table 2. To minimize the differences, the study was done in homogeny groups for demographical data (age, sex) and for lesion characteristic (level, severity and duration of SCI) and for the clinical assessments (GRASSP, BBT, SCIM-III, different type of MVC), and randomization allocation was used to distribute the subject to the experimental groups.
Point 16:Why did you not consider intention to treat analysis? Intention-to-treat analysis is a method for analyzing results in a prospective randomized study where all participants who are randomized are included in the statistical analysis and analysed according to the group they were originally assigned, regardless of what treatment they received
Response 16: Thank you very much for the suggestion and we consider that Intention-to-treat analysis (ITT) could be very useful for the future studies. But here we discussed with our statisticians and after the literature' research of "intention-to-treat "ITT"" analysis whichincludes every subject who is randomized according to randomized treatment assignment, we decided that ITT could not be considered for this study because of crossover of 8 subjects.
References:
Gupta SK. Perspect Clin Res.2011 Intention-to-treat concept: A review.
Peduzzi P, et al. J Thorac Cardiovasc Surg1991; Intent-to-treat analysis and the problem of crossovers. An example from the Veterans Administration coronary bypass surgery study
Discussion
Point 17: Discuss limitations of the study, taking into account sources of potential bias or imprecision. Discuss both direction and magnitude of any potential bias.
Point 18: Give a cautious overall interpretation of results considering objectives, limitations, multiplicity of analyses, results from similar studies, and other relevant evidence
Response 17 and 18: "They are several limitations of this study: (i) We had a limited staff numbers to realize "blinding" due to COVID pandemic and the blindness of SCI subjects was not performed since they were aware if stimulation was on because during EMG recordings, they received the stimulation. (ii) Eight subjects were crossover the group: five from control to intervention and three from intervention to control group. We considered that those subjects as a new candidate because of a least 4 weeks or more time of break between each experiment. Five subjects from control group who had a less time since SCI before favouring more the change score, but the change score was higher for intervention group. iii) The study was done in a SCI population at a subacute stage, so there is a chance for neurological improvement. Both groups were homogeneous in terms of severity and time since injury in order to reduce variability."
Point 19: Discuss the generalizability (external validity) of the study results
Response 19: We added in the conclusions: “Challenges that remain include exploring the optimal dose and spinal cord segments for stimulation, timing of stimulation, and developing better strategies for tSCS according to the characteristics of the subject. For this, more studies are needed to improve implementation of tSCS, study randomization and translation to the clinical setting in big number of SCI subjects. ”
Round 2
Reviewer 1 Report
The authors have made numerous changes to the text, including stating that this is a mix between parallel group and crossover study.
However, they have not addressed the fundamental issues.
Crossover studies, especially with a multiweek intervention, cannot simply be assumed to have independent arms. There needs to be justification of the length of washout. There must be statistical analysis of carryover effects. There must be more complicated statistical analysis because there is more correlation between the groups compared to a parallel design. In short, a statistician should participate in this study’s analysis – and in peer review.
Performing a crossover study is especially problematic because of the ongoing improvement during the subacute period. The authors’ response essentially denies this. They don’t list the actual number of weeks between interventions for the 8 people who did both.
Furthermore, the answer to the critique about left and right hand is not sufficient. Of course cervical SCI is usually asymmetric. That wasn’t my point! But across a group of people with cervical SCI, there is no reason to think that left and right hands will react differently to an intervention that targets both arms. Therefore, there is no valid reason to report the left and right results separately. The only reason is that – by chance – one side had a statistically significant improvement and one didn’t.
Author Response
Thank you for the positive and constructive comments of the reviewer 1.
The authors have made numerous changes to the text, including stating that this is a mix between parallel group and crossover study.
However, they have not addressed the fundamental issues.
Point 1: Crossover studies, especially with a multiweek intervention, cannot simply be assumed to have independent arms. There needs to be justification of the length of washout.
Response 1:We added this information now as suggested: The mean washout period from the intervention to control group (70.7 ± 35.9 days, range: 42-111 days) was larger than the washout period from the control to intervention group (46.4 ± 21.6 days, range: 30-79 days). This information was added in the text.
As seen, the washout period of intervention group to control group was enough large to considered and those subjects as a new subjects. And washout period of the control group to intervention group was at least one month where we did not wait any carryover effect, because both group contained this condition.
Additionally, to further support the absence of carryover effects, statistical analysis was performed to compare the post-experiment data with the follow-up period data for both groups (5 subjects who switched from control to intervention group and 3 subjects who switched from intervention to control group). The analysis showed that there were no significant changes, as seen in the data presented in the attached table . This lack of change during the follow-up period suggests that there was no carryover effect during follow-up period, which was 2 weeks, and it is very difficult to expect any additional effect of any experimental condition, as noted by the reviewer.
OUTCOME MEASURES |
|
Intervention group (n=3) |
Control group (n=5) |
|
|
Time |
p value post-follow up |
p value post-follow up
|
|
GRASSP strength (R+ L)
|
Post Follow-up
|
1,000 |
0,180 |
|
GRASSP Sensation (R+L)
|
Post Follow-up
|
0.317
|
1,000
|
|
GRASSP prehension ability: cylindrical grasp (R + L)
|
Post Follow-up |
0,180 |
1,000 |
|
GRASSP prehension ability: lateral key pinch (R + L)
|
Post Follow-up |
0,317 |
1,000 |
|
GRASSP prehension ability: tip to tip pinch (R + L) |
Post Follow-up |
0,317
|
1,000
|
|
GRASSP total prehension ability (R + L) |
Post Follow-up
|
0,180
|
1,000
|
|
GRASSP prehension performance (R + L)
|
Post Follow-up
|
0,257
|
0,713
|
|
GRASSP (R + L)
|
Post Follow-up
|
0,357 |
0,102 |
|
BBT (R+ L)
|
Post Follow-up |
0,362 |
0,432 |
|
MVC: cylindrical grasp force (R + L)
|
Post Follow-up
|
0,713
|
0,273
|
|
MVC: lateral pinch force (R + L)
|
Post Follow-up |
0,655 |
0,285
|
|
MVC: tip to tip pinch force (R + L) |
Post Follow-up
|
0,357
|
0,102
|
|
Upper motor extremity AIS
|
Post Follow-up |
0,156 |
0,200
|
|
Total Motor Score AIS |
Post Follow-up |
0,356 |
0,200 |
|
Total Sensitive Score AIS
|
Post Follow-up
|
0,356
|
0,200
|
Point 2:There must be statistical analysis of carryover effects. There must be more complicated statistical analysis because there is more correlation between the groups compared to a parallel design. In short, a statistician should participate in this study’s analysis – and in peer review.
Response 2:We state that our subject’s crossover can be considered a new subject after a washout period, which we gave now in the text separately. And that we have compared post-evaluation values with follow-up in both groups and found no differences. This suggests that the washout period effectively removed any carry-over effects from previous experimental conditions.
Additionally, to further support the absence of carryover effects, statistical analysis was performed to compare the post-experiment data with the follow-up period data for both groups (5 subjects who switched from control to intervention group and 3 subjects who switched from intervention to control group). The analysis showed that there were no significant changes, as seen in the data presented in the table. This lack of change during the follow-up period suggests that there was no carryover effect during follow-up period and it is very difficult to expect any additional effect of any experimental condition in this study.
The data and statistical analysis were performed under the guidance of several statisticians. Furthermore, the presence of two co-authors, Kumru H and Castillo Y, has a background in statistics and bioengineering, respectively. Kumru H has a background in statistics and Castillo Y has expertise in Bioengineering, which supported the proper analysis of the data.
Point 3:Performing a crossover study is especially problematic because of the ongoing improvement during the subacute period. The authors’ response essentially denies this. They don’t list the actual number of weeks between interventions for the 8 people who did both.
Response 3:We apologize for missing the reviewer's suggestion regarding the washout period in the first revision. We have now included the information in terms of days. If the reviewer prefers it expressed in weeks, we can make that adjustment.
Point 4:Furthermore, the answer to the critique about left and right hand is not sufficient. Of course cervical SCI is usually asymmetric. That wasn’t my point! But across a group of people with cervical SCI, there is no reason to think that left and right hands will react differently to an intervention that targets both arms. Therefore, there is no valid reason to report the left and right results separately. The only reason is that – by chance – one side had a statistically significant improvement and one didn’t.
Response 4:We conducted clinical assessments for both hands and believe that including this information could be useful for researchers and physicians in clinical practice (if there is similar effect of tSCS, etc.). However, in the results section, only the data for both hands was discussed.
Presenting detailed information highlights the complexities of the study process and provides valuable insights in the results. We are uncertain as to why they should be removed.
If the editor deems it necessary, this information can be removed from the text.

Reviewer 3 Report
The authors have addressed all my comments. The manuscript can be accepted in its current form.
Author Response
Thank you for accepting the manuscript
Round 3
Reviewer 1 Report
The authors have made more clarifications in this version. Thank you.